# Taxonomic signatures of cause-specific mortality risk in human gut microbiome

Aaro Salosensaari [1,2,3,12], Ville Laitinen [2,12], Aki S. Havulinna [4,5], Guillaume Meric [6,7], Susan Cheng [8,9], Markus Perola[4], Liisa Valsta[4], Georg Alfthan[4], Michael Inouye[6,7], Jeramie D. Watrous[10], Tao Long[10], Rodolfo A. Salido[11], Karenina Sanders[11], Caitriona Brennan[11], Gregory C. Humphrey[11], Jon G. Sanders [11], Mohit Jain[10], Pekka Jousilahti[4], Veikko Salomaa [4], Rob Knight [11], Leo Lahti [2,12✉] & Teemu Niiranen[1,4,12✉]

The collection of fecal material and developments in sequencing technologies have enabled standardised and non-invasive gut microbiome profiling. Microbiome composition from several large cohorts have been cross-sectionally linked to various lifestyle factors and diseases. In spite of these advances, prospective associations between microbiome composition and health have remained uncharacterised due to the lack of sufficiently large and representative population cohorts with comprehensive follow-up data. Here, we analyse the long-term association between gut microbiome variation and mortality in a well-phenotyped and representative population cohort from Finland ($n = 7211$). We report robust taxonomic and functional microbiome signatures related to the Enterobacteriaceae family that are associated with mortality risk during a 15-year follow-up. Our results extend previous cross-sectional studies, and help to establish the basis for examining long-term associations between human gut microbiome composition, incident outcomes, and general health status.

[1] Division of Medicine, Turku University Hospital and University of Turku, Turku, Finland. [2] Department of Computing, University of Turku, Turku, Finland. [3] Department of Mathematics and Statistics, University of Turku, Turku, Finland. [4] Finnish Institute for Health and Welfare, Helsinki, Finland. [5] Institute for Molecular Medicine Finland, FIMM-HiLIFE, Helsinki, Finland. [6] Cambridge Baker Systems Genomics Initiative, Baker Heart and Diabetes Institute, Melbourne, VIC, Australia. [7] Cambridge Baker Systems Genomics Initiative, Department of Public Health and Primary Care, University of Cambridge, Cambridge, UK. [8] Division of Cardiology, Brigham and Women's Hospital, Boston, MA, USA. [9] Cedars-Sinai Medical Center, Los Angeles, CA, USA. [10] Departments of Medicine and Pharmacology, University of California San Diego, San Diego, CA, USA. [11] Department of Pediatrics, University of California San Diego, San Diego, CA, USA. [12] These authors contributed equally: Aaro Salosensaari, Ville Laitinen, Leo Lahti, Teemu Niiranen. ✉email: leo.lahti@utu.fi; teemu.niiranen@utu.fi

Following the advances in measurement technologies, microbiome composition has data from several large cohorts that have been cross-sectionally linked to various lifestyle factors and diseases[1–5]. In spite of these advances, prospective associations between microbiome composition and health have remained uncharacterised due to the lack of sufficiently large and representative population cohorts with comprehensive follow-up data[6–8].

The long research tradition in population-level health surveys, high participation rates and the availability of comprehensive, nationwide health registers that allow monitoring of health variations across an individual's lifespan have brought Finland to the forefront of population-based cohort studies[9–12]. Here, we analyse the faecal microbiome composition in a representative random sample of 7211 adults (mean age 49.5 years, 55.1% women) who participated in the FINRISK 2002 health examination survey, which included stool sample collection and cross-sectional phenotyping in 2002[9]. Access to electronic health registers and death certificates across the 15-year time span following sample collection is a unique feature of this study, allowing us to complement the earlier cross-sectional studies by associating gut microbiome profiles with a long-term follow-up of health status and mortality after the baseline examination. Here, we demonstrate that taxonomic and functional microbiome signatures related to the Enterobacteriaceae family are associated with mortality risk in the general population over an extended follow-up.

## Results

**Study sample and microbiome taxonomic composition**. Altogether, 729 of the 7055 participants (10.2%) with complete data available died during a median follow-up of 14.8 years (Fig. 1a, b). We investigated links between mortality and the key features of microbiome composition, including alpha and beta diversity, genus abundances, taxonomic co-occurrence networks and functional predictions. We observed altogether 51 phyla and 1754 genera in this cohort (Supplementary Data 1 and 2). Inter-individual variation in taxonomic composition was largely attributable to differences in the relative abundances of the most prevalent and abundant genera (Fig. 1c, Supplementary Fig. 1 and Supplementary Data 2). Most genera were rare and observed in <1% of the study participants. In addition to analysing the overall species diversity, we have focused on the 91 genus-level taxonomic groups that were detected in >1% of the study participants at a within-sample relative abundance of >0.1%. These included mostly bacterial genera (87) but also viruses (1) and archaea (3) (Supplementary Data 2), with a median relative abundance of 99.3%.

**Microbiome features and mortality risk**. We performed a prospective analysis by examining how microbiome features were related to mortality risk in a 15-year follow-up. Alpha diversity was not significantly associated with mortality (false discovery rate (FDR)-corrected $P = 0.17$, two-tailed Wald test for Cox regression coefficient, Supplementary Table 1). However, we detected a robust and significant signal between beta diversity, or the overall community variation and elevated mortality risk. We used the standard centred log-ratio (CLR) transformation to reduce compositionality bias in taxonomic abundance data and observed that the third principal component of the CLR-transformed species abundance matrix (PC3) was strongly linked to all-cause mortality risk (hazard ratio (HR), 1.14; 95% confidence interval (CI), 1.07–1.23; FDR-corrected $P = 0.001$, two-tailed Wald test for Cox regression coefficient, Fig. 2 and Supplementary Table 1). The observation was robust to factors known to affect microbiome composition and mortality risk, i.e., age, sex, BMI, smoking, diabetes, use of antineoplastic or immunomodulating agents, systolic blood pressure and use of antihypertensive medication. Surprisingly, PC3 demonstrated an even stronger association with mortality than systolic blood pressure, which has been established as the leading cause of global burden of disease (Supplementary Table 2)[13]. Moreover, the findings related to PC3 could be observed in independent samples of the Eastern and Western Finnish populations whose genetic backgrounds, lifestyles and life expectancies differ (Supplementary Fig. 2)[14,15]. PC3 was driven by species of the Enterobacteriaceae family that are a part of the normal gut microbiome but can also cause infectious diseases in the gut and other body sites Supplementary Figs. 3 and 4 and Supplementary Data 3)[16]. Inclusion of a Healthy Food Choices Score, representing a healthy diet (Supplementary Methods), among the covariates did not materially change the association between PC3 and mortality (HR, 1.13; 95% CI, 1.04–1.23; $P = 0.005$). The Healthy Food Choices Score was related to reduced mortality risk in the same model (HR, 0.86; 95% CI, 0.78–0.95; $P = 0.003$), demonstrating the robustness of the score. In addition, after recalculating the principal components in a sample of participants who had not used antibiotics 6 months prior to baseline, the association between PC3 and mortality remained significant (HR, 1.12; 95% CI 1.04–1.21; $P = 0.0003$).

**Enterobacteriaceae and cause-specific mortality**. In analyses for cause-specific death, increased Enterobacteriaceae abundance and PC3 were particularly strongly related to death from gastrointestinal and respiratory causes (Fig. 3 and Supplementary Fig. 5). Individuals in the fourth quartile of Enterobacteriaceae abundance and PC3 had 34% (95% CI, 9–64) and 49% (95% CI, 21–83) greater risks of death compared to participants in the first quartile, respectively. Several Enterobacteriaceae genera were also individually associated with mortality (Supplementary Table 3). The standardised CLR-transformed total Enterobacteriaceae abundance was associated with prevalent liver disease (beta, 0.33; 95% CI, 0.027–0.64; unadjusted $P = 0.03$, two-tailed Wald test). More in-depth analyses on the impact of virulence genes and disease outcomes were not possible due to the lack of statistical power as the prevalence of these genes among the FINRISK samples was low (<1%; Supplementary Data 4). These genes were mainly related to virulent strains of *Escherichia coli*.

**Taxonomic composition and mortality risk**. We then investigated the overall capacity of taxonomic composition in reflecting elevated mortality risk. We identified a significant linear and non-linear association between the abundances of 40 genera and mortality (FDR-corrected $P < 0.05$; Supplementary Fig. 6a and Supplementary Table 3). Furthermore, we applied Random Survival Forests to identify a combined taxonomic signature that has the strongest association with all-cause mortality. The top taxonomic features identified in this supervised analysis also included multiple Enterobacteriaceae genera (Supplementary Fig. 6b and Supplementary Data 5). However, community composition did not improve total mortality risk assessment compared to the eight covariates (c-statistic 0.798 for covariates versus 0.796 for covariates plus community composition; $P = 0.11$, paired $t$ test, 5-fold cross-validation). The c-statistic for the community composition alone was 0.634. As an additional analysis, we identified mortality-associated microbiome features in the Eastern population based on the Random Survival Forest model, and then tested their performance in the Western population (Table 1). Incorporating microbiome features did not significantly improve Random Survival Forest performance when compared to using

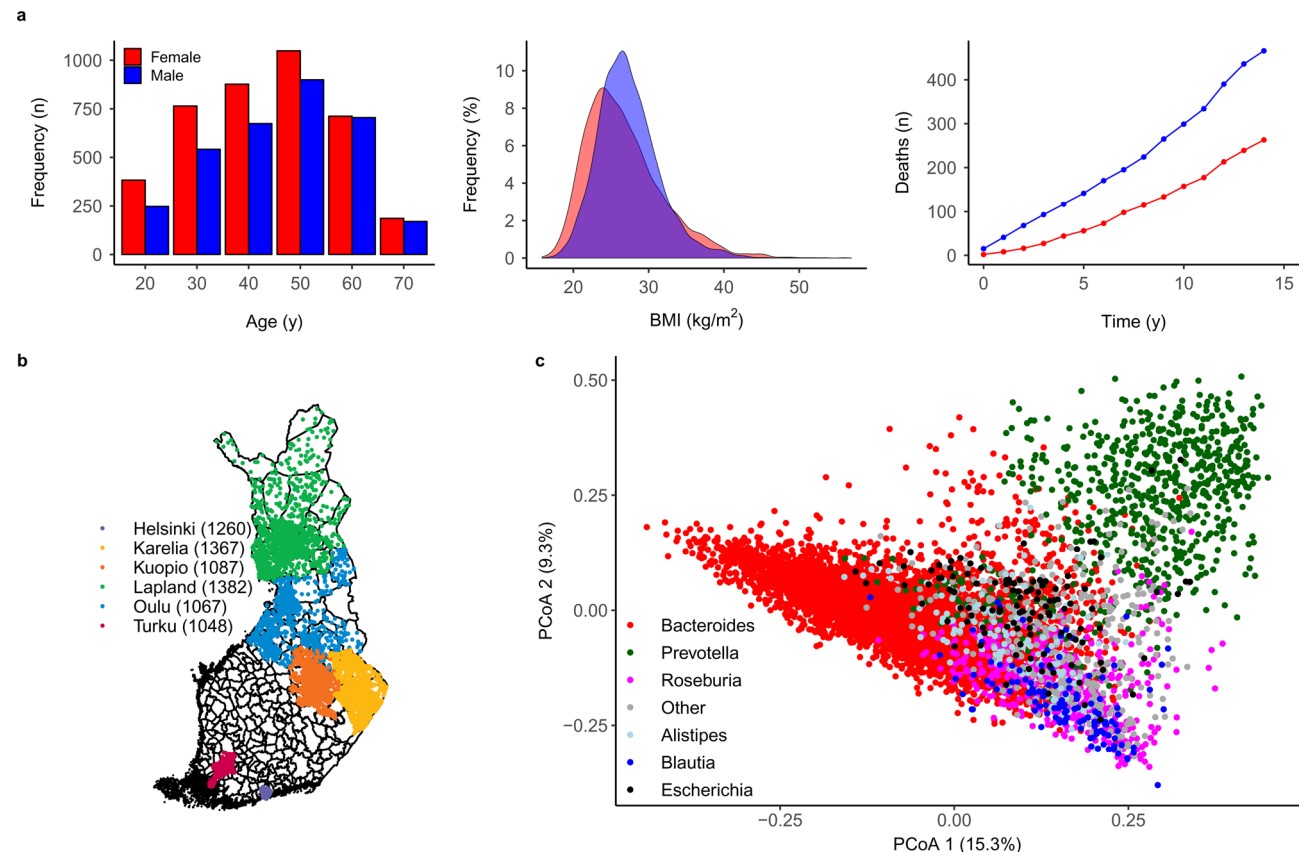

**Fig. 1 Study sample and gut microbiome characteristics. a** At baseline, the study sample (*n* = 7211) had a balanced sex ratio (55% women in red:men in blue), a mean age of 49 years (range 24–74; left panel) and a mean body mass index (BMI) of 27 kg/m² (range 16–57; middle panel). During a median follow-up time of 14.8 years, 721 of 7055 (10.2%) participants with complete data who were included in the prospective analysis died (right panel). **b** A total of 7211 out of 13,500 randomly sampled individuals (53.4% participation rate) from six catchment areas in Finland underwent stool sampling, a physical examination and filled in a questionnaire on health behaviour, history of diseases and current health. **c** Principal coordinate analysis (PCoA) indicates sample similarity based on species-level taxonomic composition. The colour indicates the dominant (most abundant) genus in each sample. Altogether, 96% of the samples are dominated by one of the six genera that are indicated in the figure.

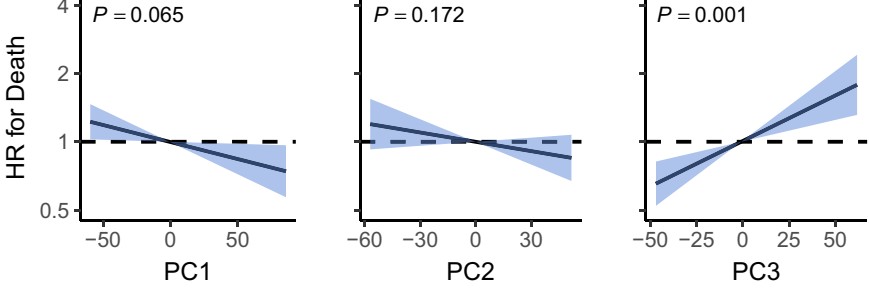

**Fig. 2 Principal components and mortality risk.** Association between mortality risk and the first three principal components of beta diversity (PC). Black line indicates the estimated hazard ratio compared to median PC value and blue area the 95% confidence interval (CI). Unit variance increase in the PCs were related to hazard ratios of 0.92 (95% CI, 0.85–0.99; FDR-adjusted *P* = 0.065; two-tailed Wald test), 0.95 (95% CI, 0.87–1.02; FDR-adjusted *P* = 0.17; two-tailed Wald test) and 1.14 (95% CI, 1.07–1.23; FDR-adjusted *P* = 0.001; two-tailed Wald test) for PC1–PC3, respectively. Analyses are adjusted for age, body mass index, sex, smoking, diabetes, use of antineoplastic and immunomodulating agents, systolic blood pressure and self-reported antihypertensive medication. The dashed line represents a hazard ratio of 1 set at median PC value. HR hazard ratio.

host covariates alone. Microbiome features were informative of mortality risk but did not bring in additional performance gains over host covariates.

**Taxonomic co-occurrence networks and mortality risk.** In order to pinpoint specific taxonomic markers that could be linked to mortality risk, we complemented the community-level analyses by shifting the focus towards a more refined sub-community

analysis. We identified groups of tightly clustered genera based on taxonomic co-occurrence network analysis. All four strongest network modules, or subnetworks (Fig. 4, Supplementary Fig. 7 and Supplementary Data 6), included genera that were linked to all-cause mortality. We observed the strongest intra-network correlations and mortality associations for the subnetwork that consisted mainly of Enterobacteriaceae genera (Fig. 4b). This subnetwork was observed in both Western and Eastern Finnish

| Cause of Death | Deaths | HR | FDR |
|---|---|---|---|
| Gastrointestinal | 36 | 1.80 (1.35–2.41) | $2.2 \times 10^{-4}$ |
| Respiratory | 31 | 1.67 (1.21–2.29) | 0.004 |
| Cancer | 238 | 1.19 (1.06–1.34) | 0.007 |
| All | 729 | 1.16 (1.09–1.25) | $9.5 \times 10^{-5}$ |
| Neurological | 60 | 1.15 (0.90–1.46) | 0.333 |
| Other | 110 | 1.10 (0.92–1.32) | 0.333 |
| Cardiovascular | 254 | 1.03 (0.91–1.16) | 0.633 |

**Fig. 3 The association between Enterobacteriaceae abundance and cause-specific mortality.** Cox hazard ratios and 95% confidence intervals are reported per unit variance increase in Enterobacteriaceae abundance. Box sizes are inversely proportional to *P* values. The entire study sample (*n* = 7211) was examined independently with each cause of death as end point. Analyses are adjusted for age, body mass index, sex, smoking, diabetes, use of antineoplastic and immunomodulating agents, systolic blood pressure and self-reported antihypertensive medication. HR hazard ratio, FDR false discovery rate.

**Table 1 Cause of death specific *c*-statistics for different feature sets.**

| Cause of death | Microbiome | Covariates | Microbiome and covariates |
|---|---|---|---|
| All | 0.63 | 0.80 | 0.79 |
| Cancer | 0.60 | 0.74 | 0.72 |
| Cardiovascular | 0.63 | 0.84 | 0.82 |
| Gastrointestinal | 0.62 | 0.63 | 0.69 |
| Neurological | 0.50 | 0.79 | 0.70 |
| Respiratory | 0.57 | 0.70 | 0.67 |
| Other | 0.60 | 0.69 | 0.76 |

Survival Random Forest was trained with the Eastern population and then tested on the Western population. A *c*-statistic of 0.5 corresponds to a random expectation, and higher values indicate improved performance. Incorporating core microbiome features did not significantly improve the *c*-statistic over the host covariates.

populations (Supplementary Fig. 8). The total abundance of this subnetwork was associated with increased mortality risk (HR = 1.16, 95% CI, 1.08–1.24; *P* = 0.0002).

**Microbiome functional profiles and mortality risk.** Using Kyoto Encyclopaedia of Genes and Genomes (KEGG) orthology (KO) groups, we assessed the potential microbial functional roles in individuals with an elevated mortality risk (Supplementary Fig. 9 and Supplementary Data 7). Many mortality-associated markers were involved in the KEGG categories related to drug biodegradation, carbohydrate metabolism, lipid metabolism and infectious diseases. These associations were both positive and negative (Supplementary Fig. 9 and Supplementary Data 7). Numerous prior studies have demonstrated that gut microbes can affect lipid and glucose metabolism and their circulating levels, which, in turn, may affect the risk of cardiometabolic disease[17,18]. Furthermore, it has been previously shown that the gut microbiota can exert direct effects on drug metabolism, potentially affecting disease risk through drug efficacy and toxicity[19]. On the other hand, several commonly used drugs have been associated with extensive changes in the taxonomy, metabolic potential and resistome of the gut microbiome[20]. Functional pathways that were negatively associated with mortality also included biological processes related to the nervous system (Supplementary Fig. 9). While these functional predictions suggest that gut–drug interactions, gut microbiome–metabolome interactions and the gut–brain axis could play a role in the development of disease, additional research is needed to confirm the drivers of the identified associations[21,22].

## Discussion

Our analysis provides a systematic quantification of the long-term health associations of the human faecal microbiome. In spite of using a remarkably heterogeneous, but robust outcome variable, we could identify specific gut microbiome features that were linked to all-cause mortality during the 15-year follow-up. These associations can be observed both in the Eastern and Western Finns who have differing genetic backgrounds, lifestyles and mortality rates[14,15]. Our results extend previously reported cross-sectional associations[1–4,23]. However, despite being a heterogeneous outcome, all-cause mortality is also a robust end point as it is virtually free of misclassification or loss to follow-up. Although individuals in the fourth quartile of PC3 had a 49% greater relative risk for all-cause mortality than those in the first quartile, microbiome signatures did not improve model discrimination. However, the *c*-statistic is a conservative method for assessing changes in model fit[24]. Even commonly accepted disease risk factors such as hypertension and smoking have only marginal impact on the *c*-statistic individually, but lead to a more accurate reclassification of large proportions of patients into higher or lower risk categories[25]. The PCA signatures are optimised to uncover maximal differences across individuals in the population, and they are thus potentially influenced by environmental and host factors. Whereas this may pose limitations for significance estimation[26,27], the unsupervised principal component analysis (PCA) has been commonly used to uncover associations between broad patterns of microbiome variation, health and environmental factors, and it provides the first step towards understanding the underlying causes. In addition, our results on the association of the gut microbiome with cause-specific mortality demonstrate that its association with some fatal outcomes is considerably stronger than with others, in spite of the lower number of events and hence reduced statistical power. The observed associations suggest that specific taxonomic configurations of the human gut microbiome may reflect health-associated changes that are linked to increased mortality, or potentially play a unique role in the maintenance of health and development of incident disease[5,16,21].

Our findings advance current research by demonstrating a particularly strong link between members of the Enterobacteriaceae family and death from gastrointestinal and respiratory causes in a general population cohort study setting with long-term follow-up. In prior cross-sectional human studies, Enterobacteriaceae have been observed to be enriched in patients with inflammatory bowel disease and colorectal cancer[28]. It has been speculated that Enterobacteriaceae, normally dominant in the upper gastrointestinal tract, become enriched in the stool due to a

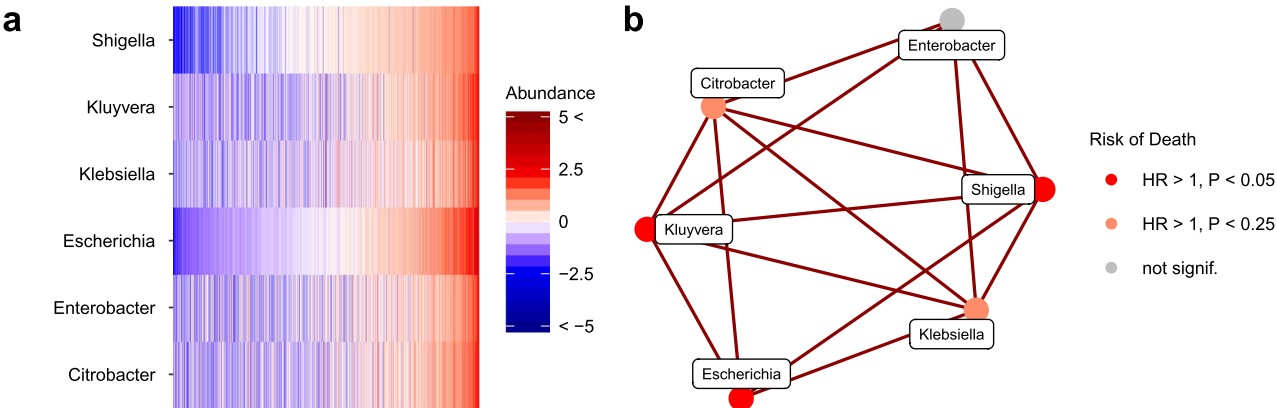

**Fig. 4 Taxonomic subnetwork associated with increased mortality risk. a** Abundance variation across the study population for the subnetwork that exhibits the strongest mortality associations (CLR-transformed abundances centred at zero and scaled to unit variance). The samples are ordered by the total relative abundance of the subnetwork. **b** The observed subnetwork structure and mortality risk. The total subnetwork abundance was associated with elevated mortality with a hazard ratio of 1.155 (95% confidence interval [CI], 1.08–1.24; $P = 0.0002$, Wald two-tailed test statistic for Cox regression, 4.07) The respective hazard ratios were 1.17 (95% CI, 1.07–1.27; $P = 0.001$, Wald statistic 3.66) in the Eastern and 1.14 (95% CI, 1.001–1.31; $P = 0.15$, Wald statistic 2.02) in the Western Finnish populations. The analyses are conducted after excluding rare taxa and adjusted for age, body mass index, sex, smoking, diabetes, use of antineoplastic and immunomodulating agents, systolic blood pressure and self-reported antihypertensive medication; $P$ values are FDR-adjusted.

faster stool transit time that occurs in diarrhoea, a symptom of many gastrointestinal diseases[29]. Host-mediated inflammation has also been shown to disrupt the gut microbiome and promote the overgrowth of Enterobacteriaceae[30]. On the other hand, increased prevalence and enhanced virulence potential of gut *E. coli* have both been linked to urinary tract infections[31]. In fact, extraintestinal *E. coli* strains often exist in the gut without consequences but have the capacity to disseminate and colonise other host niches, including the blood, the central nervous system and the urinary tract, resulting in disease[32,33]. Multidrug-resistant Enterobacteriaceae cultured from skin or blood samples have also been associated with poorer outcomes in highly selected groups of intensive care unit patients over a short-term follow-up of weeks to months[34–36]. Our results improve the understanding of the Enterobacteriaceae–mortality link in the general population. That is, we demonstrate a strong association between gut Enterobacteriaceae and fatal events in a random population sample over a 15-year follow-up, allowing for increased generalisability of our results and evidence on the long-term health associations of the gut microbiome.

A particular strength of our analysis is the availability of a random, representative population sample comprising thousands of adults from a northern European population and the access to comprehensive electronic health registers. Furthermore, our findings were supported in both Eastern and Western Finnish populations. As such, this can complement the findings from earlier cross-sectional population studies that have had a more limited representativeness based on their focus on specific populations[1,37], lack of random sampling[1–4] or low participation rates[38]. Although our sequencing depth was limited by financial constraints, it has been previously demonstrated that shallow-shotgun sequencing provides data that are strongly correlated with ultradeep-sequenced data[39]. Our data contained many rare genera and species. In order to reduce noise in taxonomic profiling and potential misclassification between closely related species that influence in particular low-abundant taxa[40] and to focus on population-level variation, we excluded the least prevalent genera from most analyses (Supplementary Table 2), and emphasise the role of Enterobacteriaceae as a family associated with elevated mortality risk. In addition, our data lacked certain covariates that have been recently linked to microbiome

composition, such as stool consistency and faecal chromogranin A[1–4]. The results of our analysis, therefore, need to be further replicated in independent cohorts with a long-term follow-up of health status. Despite the current lack of cohorts that could be used for external replication of our prospective results, our findings have implications on the design of future studies that aim to map microbiome–health associations across extended periods of time.

Until now, prospective long-term data linking microbiome composition with incident outcomes have been unavailable. Our data provide a proof of concept that the microbiome can be used to assess mortality risk, and potentially also disease risk. Additional studies will be needed to assess which disease states can be most effectively predicted through microbiome profiling. In addition, our findings can help establish a framework for recruiting disease-susceptible individuals to randomised trials to assess causal effects of gut microbiome variation on health outcomes. However, extensive research is still warranted before human microbiome sequencing can be used for prediction, prevention and targeted treatment of disease.

## Methods
**Study sample**. The FINRISK population surveys have been performed every 5 years since 1972 mainly to monitor trends in cardiovascular disease risk factors in the Finnish population. The FINRISK 2002 study was based on a stratified random sample of the population aged 25–74 years from specific geographical areas of Finland (Fig. 1)[9]. The survey included participants from North Karelia and Northern Savo in eastern Finland, Turku and Loimaa regions in southwestern Finland, the cities of Helsinki and Vantaa in the capital region, the provinces of Northern Ostrobothnia in northwestern Finland, Kainuu in northwestern Finland and the province of Lapland in northern Finland. The sampling was stratified by sex, region and 10-year age group so that each stratum had 250 participants. In North Karelia, Lapland and the cities of Helsinki and Vantaa, the strata with 65–74-year-old men and women were also sampled, each with 250 participants. The original population sample was thus 13,500 (excluding 64 who had died or moved away between sampling and the survey); the overall participation rate was 65.5% ($n = 8798$). We successfully performed stool shotgun sequencing in $n = 7231$ individuals. After excluding 20 individuals with low read counts (<50,000), $n = 7211$ participants (mean age 49.5 years, 55.1% women) remained for unsupervised analysis, of whom $n = 7055$ had the full covariate information available for survival analysis (Fig. 1). The study protocol of FINRISK 2002 was approved by the Coordinating Ethical Committee of the Helsinki and Uusimaa Hospital District (Ref. 558/E3/2001). All participants signed an informed consent. The study was conducted according to the World Medical Association's Declaration of Helsinki on ethical principles. Due to a lack of external cohorts with microbiome and long-

term mortality data, we used two internal subsamples of 4979 Eastern Finns (mainly from Northern Karelia, Northern Savo, Kainuu and Northern Ostrobothnia regions) and 2232 Western Finns (mainly from Helsinki, Turku and Loimaa regions). Altogether, 4871 and 2184 samples had complete covariate information, respectively. We used this to examine the robustness of the results in distinct subpopulations within the cohort. These two subsamples were chosen due to their well-known differences in genetic backgrounds, lifestyles and mortality rates[14,15,23]. In addition, sensitivity analyses were performed in subsamples of (1) 6191 individuals without antibiotics use 6 months prior to baseline and (2) 5727 individuals with dietary data available.

**Baseline examination.** The FINRISK 2002 survey included a self-administered questionnaire, physical measurements and collection of blood and stool samples. The questionnaire, together with an invitation to the health examination, was sent by mail to all subjects. Trained nurses carried out a physical examination and blood sampling in local health centres or other survey sites. The participants were advised to fast for ≥4 h and avoid heavy meals earlier during the day. The venous blood samples were centrifuged at the field survey sites, stored at −70 °C and transferred daily to the laboratory of the Finnish Institute for Health and Welfare. Data for physiological measures, biomarkers, dietary factors, demographic factors and lifestyle factors was collected.

**Stool sample collection.** At the baseline examination, all willing participants were given a stool sampling kit with detailed instructions. The participants mailed their samples overnight between Monday and Thursday under Finnish winter conditions to the laboratory of the Finnish Institute for Health and Welfare where the samples were stored at −20 °C. The stool samples were stored unthawed until they were transferred in 2017 to the University of California San Diego for microbiome sequencing.

**Stool DNA extraction and library preparation.** A miniaturised version of the Kapa HyperPlus Illumina-compatible library prep kit (Kapa Biosystems) was used for library generation[41]. DNA extracts were normalised to 5 ng total input per sample in an Echo 550 acoustic liquid-handling robot (Labcyte Inc). A Mosquito HV liquid-handling robot (TTP Labtech Inc. was used for 1/10 scale enzymatic fragmentation, end-repair and adapter-ligation reactions). Sequencing adapters were based on the iTru protocol[42], in which short universal adapter stubs are ligated first and then sample-specific barcoded sequences added in a subsequent PCR step. Amplified and barcoded libraries were then quantified by the PicoGreen assay and pooled in approximately equimolar ratios before being sequenced on an Illumina HiSeq 4000 instrument to an average read count of ~900,000 reads per sample.

**Taxonomic and functional profiling from sequencing data.** We analysed shotgun metagenomic sequences using a pipeline built with the Snakemake bioinformatics workflow library[32,33]. We trimmed the sequences for quality and adapter sequences using Atropos[34], and removed host reads by genome mapping against the human genome assembly GRCh38 with Bowtie2[35]. We assigned sequences taxonomy using SHOGUN v1.0.5[36] against a database containing all complete bacterial, archaeal and viral genomes available from NCBI RefSeq as of version 82 (May 8, 2017). SHOGUN calls Bowtie2 to align sequencing data against reference genomes. For each query sequence, up to 16 hits were returned in order to maximise the inclusion of closely related organisms to which the query sequence matches equally or similarly well (i.e. they all have a chance of being the true positive). As a trade-off, this behaviour could potentially result in a larger number of organisms than Bowtie2's default behaviour, which returns one hit per query. We then processed the results to estimate the relative abundance of taxa. Most genera were rare and observed in <1% of the study participants. In order to reduce the number of false positives that may contaminate low-abundant signals[40], we excluded taxonomic groups that were detected with <1% prevalence at a relative abundance of 0.1%. We excluded plasmids from all analyses. We did not rarefy the counts to the lowest sampling depth to avoid loss of data; 99.7% of the samples had a sequencing depth >50,000. Functional profiles were calculated from a combination of observed and predicted KO group annotations from the RefSeq genomes following the default parameters of the SHOGUN tool[36]. Briefly, the final KO table represents a weighted average of directly observed functional genes and those estimated to be present but unsampled based on their predicted presence within an observed genome. A full description of the method has been published[36].

**Virulence genes.** The 7211 FINRISK samples were matched to the Virulence Factor Database (VFDB; DNA sequences of the full database)[43]. Anvi'o (v5.5) was used to build a Bowtie2 database from the VFDB FASTA files and to map the FINRISK reads to the VFDB genes using the Anvi'o default setting and 99% sequence similarity[44]. The coverage was analysed with samtools (v1.9). A coverage of 500 bp and 90% of the VFDB gene length was required. The prevalence of the VFDB genes accepted with these filters is shown in Supplementary Table 7[45].

**Register linkage for pre-existing (prevalent) diseases, medication use at baseline and mortality.** In Finland, each permanent resident is assigned a unique personal identity number at birth or after immigration, which ensures reliable linkage to the computerised health registers. The nationwide Finnish health registers ensure in practice 100% coverage of all major health events (Hospital Discharge Register), all prescription drug purchases (Drug Purchase and Special Drug Reimbursement Registers) and all deaths (Causes-of-Death Register). The quality of the diagnoses in the Finnish national registers has been previously validated[10,11]. We obtained dates and causes of deaths from the National Causes-of-Death Register. The participants were followed through December 31, 2017. We observed 729 deaths between the baseline and end of follow-up period; 519 deaths occurred in Eastern Finns and 210 in Western Finns (511 and 210 with complete covariate information). The fatal events were also categorised according to their underlying cause of death (cardiovascular, 10th revision of the International Statistical Classification of Diseases code I*, n = 218; cancer, C*, n = 205; other, n = 98; neurological, G*, n = 53; gastrointestinal, K*, n = 34; respiratory, J*, n = 26).

**Covariates.** We calculated body mass index (BMI) as weight in kilograms divided by height in metres squared. Smoking (n = 1648) was defined as daily use of tobacco products. We defined diabetes (n = 401) as having a health event with an International Classification of Diseases code[46] of E10-14 in the Hospital Discharge Register or having a drug reimbursement code for diabetes in the Special Drug Reimbursement Register prior to baseline. We defined prevalent liver disease (n = 42) as a diagnosis with ICD-10 codes K70-77 at the baseline. We defined systolic blood pressure based on the mean of three measurements performed by a nurse using a mercury sphygmomanometer. n = 1096 participants self-reported antihypertensive medication use. Use of antineoplastic or immunomodulating agents (n = 62) was defined as a purchase of medications with an Anatomical Therapeutic Chemical (ATC) code of L* recorded in the Drug Purchase Register up to 4 months prior to baseline[47]. Antibiotics use (n = 1020) was defined as an ATC code of J01* up to 6 months prior to baseline. For individuals with no missing data, we defined a Healthy Food Choices Score based on a 42-item food propensity questionnaire that had choices ranging from 1 to 6 for consumption frequency (Supplementary Methods).

**Statistical methods.** Our statistical analysis workflow provides a systematic approach for microbiome-based survival analysis in prospective population cohort studies based on standard statistical techniques in microbiome bioinformatics. The workflow is described in more detail in Supplementary Methods. It was specifically designed for this study and could serve as a methodological basis for related future studies. We conducted all statistical analyses using R[48]. We standardised all phenotype variables except dichotomous variables. We controlled all Cox proportional hazards models and Random Survival Forests by including age, BMI, sex, smoking, diabetes, use of antineoplastic or immunomodulating agents, systolic blood pressure and self-reported antihypertensive medication use as covariates in the models, unless otherwise indicated. We corrected for multiple testing using FDR correction (Benjamini–Hochberg)[49]. We report the P values, where we considered an FDR-corrected P < 0.05 significant.

*Alpha diversity.* We characterised the alpha diversity of the microbiome with the Shannon index using the complete species-level abundance data.

*Beta diversity.* We used the common combination of (non-linear) principal coordinate analysis based on Bray–Curtis dissimilarity index (estimated with the R package phyloseq[50]) to visualise the overall population variation of microbiome composition. For statistical analysis of microbiome variation and mortality risk, we used the (linear) PCA based on between-samples Aitchison distances that were obtained by using the CLR-transformed abundance data (R function "prcomp"). The beta-diversity analysis was based on species-level abundance data. The first three principal components that were included in our analysis explained 3.9% (PC1), 1.6% (PC2) and 1.4% (PC3) of the observed variation.

*Taxonomic co-occurrence network detection.* After excluding the rare taxa, we detected sparse taxonomic co-occurrence (sub)networks with SPIEC-EASI[51] (R package SpiecEasi) with the (bounded) StARS model selection ("bstars"), with parameters "lambda.min.ratio", "nlambda" that determine the lambda path set to 1e − 2 and 30 following documentation recommendations. We carried out StARS with the prior stability parameter beta set at 0.01 to obtain a regularisation level that produces stable networks under subsampling, with 50 subsampling rounds ("rep.num")[52]. We excluded the subnetworks with less than three members from further analysis.

*Survival analysis.* We tested the association between the abundance of each genus, the Enterobacteriaceae family and the first three principal components with mortality using Cox proportional hazards models[53] (two-tailed Wald tests for linear or $\chi^2$ tests for non-linear associations; R package survival[54]). The package implements the Cox proportional hazards model, which gives the hazard function for subject $i$ at time $t$ the form $\lambda(t|X_i) = \lambda_0(t) \exp(\beta X_i) \exp(\beta X_i)$, where $\lambda_0(t)$ is the baseline hazard function, $X_i$ the vector of covariate variables for subject $i$ and $\beta$ are the regression coefficients

that are estimated by maximising the corresponding partial likelihood. The relative abundances were CLR-transformed in order to remove the sample comparison biases arising from unit-sum restriction of compositional data. Moreover, to recover potential non-linear associations, we modelled genus abundance both linearly and with penalised cubic splines (R function "pspline" with the default parameters; R package survival). The alpha diversity and principal components 1–3 were treated similarly. We used the Benjamini-Hochberg method to correct for multiple testing. We also reported the mortality risk ratio of individuals in the fourth quartile of PC3 relative to individuals in the first quartile in the Cox model. We assessed the proportional hazards assumption using Schoenfeld residuals. We further analysed the robustness of the observed associations by (i) rarifying the data to the lowest 10% read count, (ii) excluding the samples that belong to the lowest 10% read count quantile and (iii) without any exclusion criteria. The PC3 associations with mortality (Supplementary Data Fig. 5) remained significant after these changes (FDR < $7.5 \times 10^{-4}$) for all-cause mortality in these data subsets. Similarly, the association between the Enterobacteriaceae family and mortality (Fig. 3) remained robust to these changes (FDR < $1.7 \times 10^{-4}$)[54]. The alpha diversity and principal components 1–3 were treated similarly. We used the Benjamini–Hochberg method to correct for multiple testing. We also reported the mortality risk ratio of individuals in the fourth quartile of PC3 relative to individuals in the first quartile in the Cox model. We assessed the proportional hazards assumption using Schoenfeld residuals. We further analysed the robustness of the observed associations by (i) rarifying the data to the lowest 10% read count, (ii) excluding the samples that belong to the lowest 10% read count quantile and (iii) without any exclusion criteria. The PC3 associations with mortality (Supplementary Data Fig. 5) remained significant after these changes (FDR < $7.5 \times 10^{-4}$) for all-cause mortality in these data subsets. Similarly, the association between the Enterobacteriaceae family and mortality (Fig. 3) remained robust to these changes (FDR < $1.7 \times 10^{-4}$).

We tested the relation of the community composition with mortality using multivariate Random Survival Forest[55] (R package randomForestSRC[56]). We used default settings and measured the performance of this method with Harrell's $c$-statistic[57] in 5-fold cross-validation and then calculated the importance scores using all subjects. As an additional analysis, Random Survival Forest was trained on the Eastern and then tested on the Western population. The $c$-statistic was used as the performance metric and the model was trained and tested separately using the same three predictor sets as in the main analysis (microbiome, covariates, microbiome and covariates).

*Prevalent liver disease.* The cross-sectional association between prevalent liver disease at the time of sampling and Enterobacteriaceae was assessed by a linear model, with a CLR-transformed total Enterobacteriaceae abundance as the dependent variable and liver disease status and the aforementioned covariates as predictors.

*Functional analysis.* We associated each KO group with mortality in Cox proportional hazards models. We used $\log(1 + x)$ transformation for the KO groups to reduce skewness in the data and facilitate the use of linear models. The KO groups were then analysed using a standard linear model to determine the direction of association for all KO groups. We used the FuncTree application to analyse and visualise the functional enrichment of the gut microbiome in individuals with an increased risk of death[58]. For the module, pathway and biological process layers, we used node sizes that corresponded to the average inverse $P$ value of all KO groups that could be assigned to that node. Analyses were performed separately for KO groups that were related positively or negatively to mortality.

**Reporting summary**. Further information on research design is available in the Nature Research Reporting Summary linked to this article.

## Data availability
The metagenomic data are available from the European Genome-Phenome Archive (accession number EGAD00001007035). The phenotype data contain sensitive information from healthcare registers and they are available through the THL biobank upon submission of a research plan and signing a data transfer agreement (https://thl.fi/en/web/thl-biobank/for-researchers/application-process).

## Code availability
The source code for the analyses is available at https://doi.org/10.5281/zenodo.4306060.

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

## Acknowledgements

We thank all participants of the FINRISK 2002 survey for their contributions to this work, and Tara Schwartz for assistance with laboratory work, and Kari Koponen for assistance with the Healthy Food Choices index. This research was supported in part by grants from the Finnish Foundation for Cardiovascular Research, the Emil Aaltonen Foundation, the Paavo Nurmi Foundation, the Urmas Pekkala Foundation, the Finnish Medical Foundation, the Academy of Finland (295741, 307127 to L.L., V.L.; 321351 to T.N.; 321356 to A.S.H.), UTUGS graduate school (to V.L.) and the National Institutes of Health (R01ES027595 to M.J., K01DK116917 to J.D.W., R01HL134168, R01HL131532, R01HL143227 and R01HL142983 to S.C.). Additional support was provided by Illumina Inc. and Janssen Pharmaceutica through their sponsorship of the Center for Microbiome Innovation at UCSD.

## Author contributions

V.S., R.K., L.L. and T.N. designed the work, A.S.H., J.D.W., M.P., L.V., G.A., G.C.M., T.L., M.J., P.J., V.S., R.A.S., K.S., C.B. and G.C.H. acquired the data, A.S., V.L., L.L., J.G.S. and T.N. analysed the data and M.I., M.J., P.J., S.C., V.S., R.K., L.L. and T.N. supervised the work. All authors wrote the manuscript and gave final approval of the version to be published.

## Competing interests

The authors declare the following competing interests: V.S. has consulted for Novo Nordisk and Sanofi and received honoraria from these companies. He also has ongoing research collaboration with Bayer AG, all unrelated to this study. The other authors declare no competing interests.
