## [Peer Review File · Nature Communications]

Peer Review File

REVIEWER COMMENTS

Reviewer #1 (Remarks to the Author):

Despite the fact that there are some limitations to this study which cannot be fixed in retrospect, I nonetheless think this impressive cohort should be published in Nature Communications, even given the limitations. These limitations are now clearly stated, and the results are toned down and not over-interpreted as they were in previous versions.

Reviewer #5 (Remarks to the Author):

The response to concerns of the biostatistical reviewer is satisfactory.

Reviewer #6 (Remarks to the Author):

Overall, I find that the comments of the reviewers are well addressed and for those not discussed below, I would consider the answer satisfactory. There are however some issues that require further clarification.

- Non-replication of results when rarefying. I am well aware of the rarefaction debate and there are some things to be said for not rarefying. However, as the main association pertains a group of bacteria often associated with diarrhea and thus low read counts, the authors should be careful that the signal is not driven by the low read count samples alone and will thus have very little generalization potential. For this reason, I do think that the robustness needs to be properly tested, and results fully reported in the manuscript. I would show the following 3 analyses: (1) results as done without rarefaction; (2) results with the 10th percentile rarefaction; (3) results without rarefaction but with omission of low read count individuals (can also be 10th percentile or even multiple cutoffs), and fully discuss the effects of these tests – if 2/3 analyses are not significant, will these results ever be generalizable? With regard to the PC3 overlap problems, is a constrained PCOA not an option to look into? Also, if you say “remain qualitatively robust to rarefying” – you mean that they are not significant but just show a similar trend?

- answer 3.2 is also not satisfactory: I agree with that reviewer. It is disturbing that the main trend comes from disease groups with the smallest number of cases, yet seems to have a massive effect on the overall signal. If e.g. gastrointestinal cause of death is removed, the overall effect is probably also gone. So I agree the title should be downtoned and be more specific.

-

1. Non-replication of results when rarefying. I am well aware of the rarefaction debate and there are some things to be said for not rarefying. However, as the main association pertains a group of bacteria often associated with diarrhea and thus low read counts, the authors should be careful that the signal is not driven by the low read count samples alone and will thus have very little generalization potential. For this reason, I do think that the robustness needs to be properly tested, and results fully reported in the manuscript. I would show the following 3 analyses: (1) results as done without rarefaction; (2) results with the 10th percentile rarefaction; (3) results without rarefaction but with omission of low read count individuals (can also be 10th percentile or even multiple cutoffs), and fully discuss the effects of these tests – if 2/3 analyses are not significant, will these results ever be generalizable? With regard to the PC3 overlap problems, is a constrained PCOA not an option to look into? Also, if you say “remain qualitatively robust to rarefying” – you mean that they are not significant but just show a similar trend?

Reply: We thank the reviewer for the comments. In order to quantify robustness of the results we have re-run the main experiments concerning the association between all-cause mortality (Fig. 3) and the principal coordinate axes and cause-specific mortality (Fig. 2) with the Enterobacteriaceae family. We used the datasets proposed by the Reviewer: (1) full data, (2) data with 10th percentile rarefaction and (3) data with samples with the lowest 10th percentile read counts omitted (**Reviewer Table 1, Reviewer Fig. 1, Reviewer Fig. 2**). Conclusions of the manuscript remained qualitatively unaffected. By this we mean that the significance of the observations, and the main conclusions remain unaffected, even though the p-values and effect sizes have small variations, as expected.

The association between PC3 and all-cause mortality is statistically significant in all three variations of the data that were proposed by the Reviewer. While testing these associations, we noticed that we had mistakenly written in the earlier response that the PC3 association was not significant after rarefaction; we have now carefully checked the results and we can confirm that the reported PC3 association is significant in all three cases (full data, rarefied data, data with 10% lowest read counts excluded).

We also re-ran the Cox proportional hazard model with abundance of the family Enterobacteriaceae as predictor and cause-specific mortality as the response variable (**Reviewer Fig. 3**). Here again, the reported associations remain significant.

Reviewer Table 1. Association between PC3 and all-cause mortality using three alternative datasets.

Data	HR	P-value
Full data	1.142 (95% CI, 1.064-1.227)	7.5x10 ⁻⁴
Rarefied	1.156 (95% CI, 1.082-1.235)	4.8x10 ⁻⁵
Read count cut off	1.174 (95% CI, 1.09-1.265)	7.0x10 ⁻⁵

Reviewer Fig. 1 The association between principal coordinate axes 1-3 and all-cause mortality. Results using data with all samples (A), data with 10th percentile rarefaction (B) and data with lowest 10th percentile read count samples omitted (C).

Reviewer Fig. 2 The association between PC3 and cause-specific mortality in rarefied data. Results using data with all samples (top), data with 10th percentile rarefaction (middle) and data with lowest 10th percentile read count samples omitted (bottom).

Cause of Death	Deaths	HR	FDR
Gastrointestinal	36	1.80 (1.35-2.40)	2.2×10^{-4}
Respiratory	31	1.67 (1.21-2.30)	0.004
Cancer	239	1.19 (1.06-1.35)	0.006
All	731	1.17 (1.09-1.25)	7.1×10^{-5}
Neurological	60	1.15 (0.90-1.47)	0.330
Other	110	1.10 (0.92-1.32)	0.330
Cardiovascular	255	1.04 (0.92-1.17)	0.562

Cause of Death	Deaths	HR	FDR
Gastrointestinal	32	1.83 (1.34-2.48)	4.2×10^{-4}
Respiratory	28	1.75 (1.26-2.44)	0.002
Cancer	212	1.20 (1.06-1.36)	0.009
All	641	1.17 (1.09-1.26)	1.6×10^{-4}
Neurological	46	1.15 (0.87-1.51)	0.452
Other	101	1.07 (0.89-1.29)	0.524
Cardiovascular	222	1.04 (0.92-1.18)	0.524

Cause of Death	Deaths	HR	FDR
Gastrointestinal	32	1.78 (1.30-2.44)	0.001
Respiratory	28	1.76 (1.26-2.47)	0.002
Cancer	212	1.20 (1.05-1.36)	0.010
All	641	1.16 (1.08-1.25)	0.001
Neurological	46	1.15 (0.87-1.51)	0.463
Other	101	1.04 (0.86-1.26)	0.715
Cardiovascular	222	1.03 (0.91-1.17)	0.715

Review Fig. 3 The association between Enterobacteriaceae and cause-specific mortality. Results using data with all samples (top), data with 10th percentile rarefaction (middle) and data with lowest 10th percentile read count samples omitted (bottom).

We have revised the manuscript text as follows:

Page 18: We further analysed the robustness of the observed associations by (i) rarifying the data to the lowest 10% read count, (ii) excluding the samples that belong to the lowest 10% read count quantile, and (iii) without any exclusion criteria. The PC3 associations with mortality (**Extended Data Fig. 5**) remained significant after these changes ($FDR < 7.5 \times 10^{-4}$ for all-cause mortality in these data subsets). Similarly, the association between the *Enterobacteriaceae* family and mortality (**Fig. 3**) remained robust to these changes ($FDR < 1.7 \times 10^{-4}$; data not shown).

2. answer 3.2 is also not satisfactory: I agree with that reviewer. It is disturbing that the main trend comes from disease groups with the smallest number of cases, yet seems to have a massive effect on the overall signal. If e.g. gastrointestinal cause of death is removed, the overall effect is probably also gone. So I agree the title should be downtoned and be more specific.

Reply: We have now modified the manuscript title as suggested. The new title is:
“Taxonomic Signatures of Cause-Specific Mortality Risk in Human Gut Microbiome”

REVIEWERS' COMMENTS second round -

Reviewer #6 (Remarks to the Author):

the authors have satisfactorily addressed my concerns